Interpreting pathologies in extant and extinct archosaurs using micro-CT

Anné Jennifer 1 jennifer.anne@manchester.ac.uk
Garwood Russell J. 1
Lowe Tristan 2
Withers Philip J. 2
Manning Phillip L. 1
1 School of Earth, Atmospheric and Environmental Sciences, University of Manchester , Manchester , UK
2 Manchester X-ray Imaging Facility, School of Materials, University of Manchester , Manchester , UK
Esteban María Ángeles
Electronic publication date: 2015 Jul 28
Publication date: 2015
Volume: 3
Electronic Location ID: e1130
Received 2015 May 5; Accepted 2015 Jul 6
Copyright: © 2015 Anné et al.
Copyright year: 2015
Copyright holder: Anné et al.
License: This is an open access article distributed under the terms of the Creative Commons Attribution License, which permits unrestricted use, distribution, reproduction and adaptation in any medium and for any purpose provided that it is properly attributed. For attribution, the original author(s), title, publication source (PeerJ) and either DOI or URL of the article must be cited.
License URL: https://creativecommons.org/licenses/by/4.0/

Keywords: Palaeopathology, Micro-CT, Archosaur

Funding: The University of Manchester Dean’s Award Jurassic Foundation EPSRC EP/F007906/1 EP/F001452/1 EP/I02249X/1 This project was partially funded by the Jurassic Foundation. Jennifer Anné is funded by the Dean’s Award (University of Manchester). The MXIF facilities are partially funded by a number of EPSRC grants, including EP/F007906/1, EP/F001452/1 and EP/I02249X/1. The funders had no role in study design, data collection and analysis, decision to publish, or preparation of the manuscript.

==============================
Palaeopathology offers unique insight to the healing strategies of extinct organisms, permitting questions concerning bone physiology to be answered in greater depth. Unfortunately, most palaeopathological studies are confined to external morphological interpretations due to the destructive nature of traditional methods of study. This limits the degree of reliable diagnosis and interpretation possible. X-ray MicroTomography (micro-CT, XMT) provides a non-destructive means of analysing the internal three-dimensional structure of pathologies in both extant and extinct individuals, at higher resolutions than possible with medical scanners. In this study, we present external and internal descriptions of pathologies in extant and extinct archosaurs using XMT. This work demonstrates that the combination of external/internal diagnosis that X-ray microtomography facilitates is crucial when differentiating between pathological conditions. Furthermore, we show that the use of comparative species, both through direct analysis and from the literature, provides key information for diagnosing between vertebrate groups in the typical pathological conditions and physiological processes. Micro-CT imaging, combined with comparative observations of extant species, provides more detailed and reliable interpretation of palaeopathologies. Micro-CT is an increasingly accessible tool, which will provide key insights for correctly interpreting vertebrate pathologies in the future.

Introduction

Palaeopathology is the study of ancient diseases and trauma, and is usually limited in vertebrates to lesions that affect the skeleton, as soft tissues are generally lost over time (Rothschild & Martin, 2006; Rühli et al., 2007; Straight et al., 2009; Waldron, 2009; Rothschild & Depalma, 2013). The identification and classification of palaeopathologies can be difficult due to complex internal morphology and the loss of detail during fossilisation due to taphonomic overprint. Despite these problems, most palaeopathological studies rely on external gross morphological descriptions rather than examining internal features. Although some palaeopathologies can be identified by external examination (e.x. fracture callus) a lack of internal morphological information may lead to misidentification and over interpretation. Internal structure can be identified using thin section analysis, but this method is destructive and therefore cannot be applied to specimens that are rare or fragile. Additionally, the two-dimensional nature of this technique does not capture the entire complexity of pathological tissue unless serial sections are made. The application of medical scanners to such material has allowed for non-destructive three-dimensional investigation of internal histologic superstructures, though the resolution may be low (Straight et al., 2009; Particelli et al., 2012; Sutton, Rahman & Garwood, 2014), and medical X-ray sources struggle to penetrate dense fossils.

Since the 1990s, XMT has revolutionised the study of fine and complex bone structures in medical (Englke et al., 1999; Particelli et al., 2012), forensic (Thali et al., 2003) and palaeontological studies (Straight et al., 2009; Sutton, Rahman & Garwood, 2014; Bishop et al., 2015; Foth et al., 2015). Such scanners can attain sub-micron resolution (depending on sample size) compared to the tenths of a millimetre in conventional medical scans (Englke et al., 1999; Sutton, Rahman & Garwood, 2014). XMT is usually applied to smaller samples, as the size of the area of interest dictates spatial resolution. However, by compromising the resolution of the scan (to tens of microns versus sub-micron), this technique can be applied to larger scan areas while maintaining higher resolution scans than is possible with medical scanners (which also struggle with large specimens due to their medium energy (20–150 kV)). For example, the Nikon custom bay microtomography system in the Manchester X-ray Imaging Facility (MXIF) houses a high energy source (225–320 keV) that, coupled with a 2,000 × 2,000 pixel detector, allows decimetre-scale specimens of fossil bone to be scanned at sub-100 µm resolution. This level of detail allows us to distinguish key internal morphological features required for an accurate diagnosis of palaeopathologies. Here we present XMT data from two extant and three extinct archosaurs exhibiting a variety of pathologies, further demonstrating that XMT is a powerful tool for the field of palaeopathology.

Materials & Methods

Specimens consisted of extant and extinct archosaur material with various pathological conditions that were identified based on external observations (Table 1).

Table 1 External descriptions of specimens.

External pathological description for extant and extinct archosaurs used in this study.

Species	Skeletal element	External pathological description	
Sagittarius serpentarius (secretary bird)	pedal phalanx NHM S/1869.2.16.1	Extreme bone growth; most of the digit is obscured	
Struthio camelus (ostrich)	cervical vertebra BHI 6241	Bony growth on the postzygapothoses resulting in pseudofusion of two vertebrae	
Tyrannosaurus rex	cervical rib BHI 3033	Extensive reactive bone throughout	
Edmontosaurus annectens	metacarpal BHI 6191	Rough fracture callus caused by angular displacement	
Edmontosaurus annectens	dorsal rib BHI 6184	Large “pursed” bony growth near rib head	

Table 2 Table of scanning parameters.

Experimental parameters used for scanning with the Nikon Metris Custom Bay (MXIF).

Species	kV	µA	Filter (mm)	Exposure (ms)	Voxel size (µm)	Total scan time (min)	
Sagittarius serpentarius	50	150	none	1,000	16.6	55	
Struthio camelus	60	170	1.5 Al	1,000	44.9	55	
Tyrannosaurus rex	60	170	1.5 Al	1,000	15.3	55	
Edmontosaurus annectens—metacarpal	115	115	0.25 Cu	500	47.9	30	
Edmontosaurus annectens—rib	115	85	0.25 Cu	708	78	40	

Extant taxa were included to improve palaeopathological diagnosis as medical terminology, physiology of skeletal elements, and likelihood of certain diseases can differ, even between human and mammalian companion animals (Huchzermyer & Cooper, 2000; Rothschild & Panza, 2005; Mader, 2006; Rothschild & Martin, 2006; Kranenburg, Hazewinkel & Meij, 2013; Foth et al., 2015).

XMT scanning was conducted at the Manchester X-ray Imaging Facility (MXIF) using the Nikon Metris Custom Bay, a system which can accommodate large specimens (maximum field of view of 410 mm) and provide spatial resolutions between 100 and ∼3.5 µm (Table 2). The system has a 225 kV static multi-metal anode target, which was set to tungsten in order to achieve the maximum energy, to improve X-ray penetration of the scanned material (fossil and extant bone). The source voltage was set to 225 kV, and auto conditioned for 30 min prior to scanning to decrease the likelihood of the source cutting out while scanning. Specimens were mounted on a manipulation stage using a variety of plates and clamps depending on their size, weight, and geometry. Previous scans of this type of material have demonstrated that 15–20% transmission through the specimen relative to the flat field provide excellent scans. To achieve this aim, the source voltage and current were modified coupled with changing thicknesses of Cu or Al filters to minimise beam hardening but achieve an optimal spectral width and intensity. Exposure was selected to minimise scan times while collecting clean data (between 0.5 and 1.0 s exposure, sample dependent), and for all scans counts on the detector panel outside the sample were kept below 65,000 counts for the selected gain, which was set to minimum to reduce noise. For each sample, scanning parameters were selected to fit these requirements while using the lowest voltage possible to maximise the attenuation from photoelectric absorption and thus maximise contrast from compositional differences in the sample (Sutton, Rahman & Garwood, 2014). 3,142 projections were collected; a number based on the optimise option for the CT Pro acquisition software.

Volumes were reconstructed using the Nikon CT Pro software. Processed scans were converted to TIFF stacks using the HMtool in MATLAB®, which is an in-house script used by the MXIF. TIFFs or VGI/VOL files (the former for Volume Graphics’ VGStudio MAX) were opened using Fiji (Schindelin et al., 2012) for initial analysis of slice stacks. TIFFs were then opened in Avizo® to construct orthoslices, while VGI files were opened in Drishti for volume rendering (Limaye, 2006).

Results and Discussion

NHM S/1869.2.16.1: Extensive bone growth persists through the interior of the S. serpentarius phalanx. In many locations, this makes distinguishing normal cancellous struts from pathological growth difficult (Fig. 1). Despite the degree of pathological intrusion, both articular surfaces maintain shape and texture. A large, circular lesion is located on the plantar surface, with signs of necrosis internally. The concentric ring appearance within the necrotic area matches the description for a fibricess; a localised inflammatory process caused by the incomplete elimination of pathogens in archosaurs (Harmon, 1998; Huchzermyer & Cooper, 2000). The most likely cause is osteitis (inflammation of the bone by infection) or osteomyelitis (inflammation of bone marrow by infection; Ritchie, Harrison & Harrison, 1994; Berners, 2002). This diagnosis is based on the lesion on the plantar surface of the bone (Fig. 1E) and internal necrosis (Figs. 1B–1D). In avians, bacterial osteomyelitis is identified based on severe necrosis, with minimal periosteal reaction (Ritchie, Harrison & Harrison, 1994). However, periosteal change can occur in chronic infections, and in fungal osteomyelitis, the periosteal reaction is pronounced (Ritchie, Harrison & Harrison, 1994). Additionally, there seems to be discrepancies between veterinary diagnoses as some characterise osteomyelitis as having a pronounced periosteal reaction (Doneley, 2011).

Figure 1 S. serpentarius (NHM S/1869.2.16.1) pedal phalanx; photograph of the specimen in plantar view (A), XMT slices in medial-lateral (B), dorsal-ventral (C) and transverse (D) views, and 3D rendering of the plantar (E) and distal (F) surfaces.

A large, circular lesion is seen on the plantar surface (red arrows; A, E), with small necrotic spaces persisting throughout the phalanx (red arrows B, C). Extensive reactive bone growth persists both internally and externally. The outline of the normal bone cortex is barely visible in some areas and indistinguishable in others (red circles; D, F). The extent of the growth makes it difficult to identify any possible indicators of trauma. Both articulation surfaces are relatively untouched. Scale bar is 1 cm.

Due to the extent of the new growth, it is impossible to distinguish if the infection was a result of a fracture, or another complication such as ulcerative pododermatitis (bumblefoot; Herman, Locke & Clark, 1962; Keymer, 1972; Remple & Al-AShbal, 1993; Ritchie, Harrison & Harrison, 1994; Gentz, 1996; Huchzermyer & Cooper, 2000; Berners, 2002; Wyss et al., 2015), which can cause osteomyelitis in later stages. Pododermatitis is caused by a number of bacteria and is more common in captive individuals with poor perching surfaces. However, it has been documented in wild individuals, usually as a result of a puncture (Herman, Locke & Clark, 1962; Keymer, 1972; Remple & Al-AShbal, 1993; Gentz, 1996; Berners, 2002; Wyss et al., 2015). Additionally, most studies on pododermatisis concerning birds of prey focus on species that hunt ‘on the wing.’ Secretary birds spend most of their time foraging on hard ground, using their feet to stamp their prey. Thus pododermatisis is a reasonable hypothesis for the cause of osteomyelitis.

Excluded conditions include gout, osteopetrosis and neoplasia. Gout is a common metabolic disease in archosaurs caused by concentration of urate crystals (Berners, 2002; Mader, 2006). Although common in wild birds (Morishita, Aye & Brooks, 1997), especially within the pedal phalanges, gout affects the joint surfaces, which in this specimen are unaffected (Mader, 2006). Osteopetrosis causes thickening of the bone through prolific bone deposition, resulting in the loss of the medullary cavity (Ritchie, Harrison & Harrison, 1994; Doneley, 2011); however, this is has only been noted in the femur, ulna, radius, pectoral girdle and vertebrae (in captive birds). Neoplasia is the most likely of the alternative, as it resembles osteomyelitis; however neoplasia is rare in wild individuals (Ritchie, Harrison & Harrison, 1994).

BHI 6241: The outer appearance of the S. camelus cervical matches descriptions for early stages of Diffuse Idiopathic Skeletal Hyperostosis (DISH). DISH is described (externally) as fused vertebrae with a ‘melted candle wax’ appearance, where fusion may be asymmetrical and the articulated surfaces appear unaffected (Fig. 2A; Rothschild & Martin, 2006; Waldron, 2009). However, DISH has not been found in avians. Newcastle disease is a common ailment of captive ratites resulting in the inability for the individual to lift their head (Stewart, 1994; Huchzermyer, 2002). However, this is a neurological disease and does not affect the bone. Alternative conditions that affect vertebrae in avians include arthritis, osteopetrosis (viral infection in avians) and vertebral osteomyelitis (bacterial or fungal) (Julian, 1998; Stalker et al., 2010; Doneley, 2011).

Figure 2 S. camelus (BHI 6241) cervical vertebra; photograph of the specimen in medial-lateral view (A) and XMT slices in medial-lateral (B), dorsal-ventral (C) and transverse (D) views.

The affected zygapophysis shows large necrotic cavities (red arrows) surrounded by relatively dense reactive bone, which spreads both internally and externally to form osteophyte ‘hooks.’ Scale bar is 1 cm.

Internal inspection provided further evidence for infection of the zygapophysis due to the presence of lesions surrounded by densely compact bone as compared to the normal zygapophysis (Figs. 2B–2D). Reactive arthritis has been identified in avians, but within long bone joints (MacLean et al., 2013), and the lack of soft tissue makes it difficult to diagnose in this specimen. Osteopetrosis is characterised by the proliferation of porous subperiosteal bone, which is not seen in BHI 6241 (Ritchie, Harrison & Harrison, 1994). Osteomylitic infection of the vertebral column has been noted in broiler chickens; however, cases affect the centrum and not the processes as seen in this specimen (Julian, 1998; Stalker et al., 2010). In snakes, vertebral osteomyelitis can cause anklyosing (fusion) of the vertebrae similar to what is seen in BHI 6241 (Stacy & Pessier, 2007). Although this is a squamate comparison, the description is the closest to what is seen in BHI 6241. Thus, the suggested diagnosis is partial ankylosing due to a form of vertebral osteomyelitis.

BHI 3033: The T. rex cervical vertebrae associated with this rib exhibit severe reactive bone growth, most likely due to a complication of healing after trauma, which caused two of the cervicals to become fused (Larson & Donnan, 2002; PL Larson, pers.comm., 2013). The complication derived from the vertebral injury appears to have spread to the cervical rib, giving it a frothy appearance, and in some areas it is enlarged (Fig. 3). The most likely cause is osteitis or osteomyelitis as a result of complications during healing.

Figure 3 T. rex (BHI 3033) cervical rib; photograph of the specimen in rostral-caudal view (A), XMT slices in medial-lateral (B), rostral-caudal (C) and transverse (D) views, and 3D rendering in medial-lateral view (E).

Reactive bone is observed in some concentrated areas (red arrows). The high porosity consists of long canals running parallel to the long axis of the specimen (E in yellow). Scale bar is 5 mm.

Internal examination reveals high levels of porosity, with canals lined parallel to the long axis of the rib, which is the normal condition (Figs. 3B, 3C and 3E). There are no signs of necrosis to suggest an exudative reaction like that seen in bacterial infection (Huchzermyer, 2003); however, there is reactive bone growth present towards the distal end of the section (red arrows). The fusion of the cervical vertebrae associated with this rib suggests a similar condition as seen in BHI 6241, where an infection of the cervicals results in inflammation of the bone/marrow cavity and anklyosing of the vertebrae. Although osteomyelitis does not always result in periosteal new bone growth in birds and reptiles, it can cause anklyosing between affected vertebrae similar to what is seen in the associated cervicals (Mader, 2006). Other conditions that could cause bone inflammation are not applicable as they typically affect the joints (gout, arthritis; Berners, 2002; Mader, 2006) or are rare in ribs (osteopetrosis; Ritchie, Harrison & Harrison, 1994). However, as osteomyelitis is difficult to diagnose in extant reptile using X-ray techniques (such as radiographs), we can only tentatively diagnosis this as a form of osteomyelitis.

BHI 6191: The E. annectens metacarpal displays a rough fracture callus (poorly remodelled) that surrounds a badly displaced fracture (Fig. 4). The pathological tissue is very porous and includes several large lesions continuing past the callus and through the metacarpal (Figs. 4B and 4C). The original morphology of the metacarpal can be seen in the transverse view (Fig. 4D) including some laminar histological features, though there is severe angular compaction and displacement (Figs. 4B and 4D). The metacarpal becomes increasingly hard to discriminate from the internal pathological growth moving distally from the apparent fracture plane, with the bone’s ends completely encompassed into the pathological mass (Figs. 4B and 4C). The diagnosis for this pathology is osteomyelitis caused by fracture complications based on the misalignment of the fractured pieces, internal necrosis and islands of ‘normal’ tissue (Rothschild & Martin, 2006; Stacy & Pessier, 2007; Gál, 2008; Waldron, 2009).

Figure 4 E. annectens (BHI 6191) metacarpal; photograph of the specimen in medial-lateral view (A) and XMT slices in medial-lateral (B), dorsal-ventral (C) and transverse (D) views.

A fracture probably caused by crushing is seen in the centre of the element, with severe angular misalignment (B). Reactive bone persists throughout the entire metacarpal, with a large rough fracture callus (poorly remodelled). Several necrotic areas are seen throughout the specimen (red arrows B, C). The ends of the metacarpal are almost completely resorbed and replaced with reactive bone (C). Despite the extent of resorption and reactive bone growth, some of the original laminar features can still be seen (red circle D). Scale bar is 1 cm.

Other common degenerative conditions known to affect archosaur digits include arthritis, gout, neoplasia, and fibrous osteodystrophy (Mader, 2006). Arthritis and gout, which are the most common of these conditions (within archosaurs), can be excluded as both affect the joints (Mader, 2006). Fibrous osteodystrophy is known to weaken long bones, increasing the occurrence of fractures, as well as cause massive bone turnover (Mader, 2006). However, it is also marked by thinning of bone tissue, which is not seen in BHI 6191. Neoplasia could be an alternative diagnosis, as it is known to cause both reactive bone growth and destroy original bone tissue (Mader, 2006; Doneley, 2011). However, as cancers are fairly rare in wild archosaurs (Effron, Griner & Benirschke, 1977; Siegfried, 1982; Garner, Hernandez-Divers & Raymond, 2004; Rothschild & Martin, 2006), we maintain the diagnosis as osteomyelitis caused by complications of a fracture.

BHI 6184: Externally, the exostosis expands radially from the E. annectens rib with no distinguishable boundary between the pathological and normal tissues (Fig. 5). The ‘pursed’ external morphology is seen internally as a secondary protrusion in medial-lateral view (Fig. 5C) and as a simple outgrowth of reactive bone in dorsal-ventral and transverse views (Figs. 5B and 5D). The original hypothesis suggested that the growth of bone occurred around an embedded foreign object such as a tooth, which has been seen in other hadrosaurians (DePalma et al., 2013). However, there is no indication here of an embedded fragment.

Figure 5 E. annectens (BHI 6184) dorsal rib; photograph of the specimen in rostral-caudal view with magnified image of the ‘folded tissue’ (A) and XMT slices in rostral-caudal (B), medial-lateral (C) and transverse (D) views.

The reactive bone growth is localised to one side of the rib (red boxes B, D). There are no signs of trauma, though smaller fractures may be concealed within the pathological mass. The “folded” morphology of the pathological mass is seen as an outgrowth of bone (red arrow C). Scale bar is 1 cm.

Another possibility is periostitis, which results in irregular periosteal reactive bone growth (Waldron, 2009). Periostitis can be remodelled, giving a smooth outside texture with time; however, in this example there is no definition between the pathological growth and normal bone surface as expected in periostitis (Figs. 5B–5D). Soft bone diseases such as rickets/osteomalacia can also cause such deformities. Osteomalacia is the softening of the bones seen in young crocodilians as a result of an inability to ossify osteoid, and is usually caused by poor calcium intake (Huchzermyer, 2003; Waldron, 2009). However, the lack of preservation of osteoid in fossilised tissues makes it difficult to assess osteomalacia in the fossil record (Rothschild & Martin, 2006). Finally, the protrusion could be a fracture callus as reptile rib fractures usually show no fracture line, but rather an increase in rib diameter (Mader, 2006). However, the other characteristics of reptilian rib fracture, such as a thinning of the cortex and widening of the medullary cavity is not observed (Mader, 2006). Therefore, we cannot diagnose the pathological condition beyond an abnormal growth of folded ossified tissue.

Conclusions

Although palaeopathological interpretation is restricted, to a degree, due to loss of information during fossilisation, detailed internal microstructural information can drastically improve the characterisation of pathological tissues. High-resolution CT (specifically XMT) provides a non-destructive means to view and aid in the diagnosis of complex internal morphologies of paleopathologies. In this study, XMT revealed fine detail morphological features that were necessary for a more informative diagnosis, including the correction of misinterpretations (ostrich, Edmontosaurus rib). Thus, for future studies, we suggest the application of both internal and external morphological descriptions when diagnosing palaeopathologies through the use of X-ray microtomography, especially when the application of thin section analysis is either not available or possible (due to rarity of sample).

We would like to thank the Black Hills Institute and the Natural History Museum (Tring) for the loan of specimens, the reviewers for the insightful comments, and the Manchester X-ray Imaging Facility.

Additional Information and Declarations

Competing Interests

Author Contributions

Russell Garwood is an 1851 Royal Commission Research Fellow and a Scientific Associate at the Natural History Museum, London. Phillip Manning is a Research Associate at the American Museum of Natural History, New York. All authors are members of the Interdisciplinary Centre for Ancient Life (ICAL).

Jennifer Anné conceived and designed the experiments, performed the experiments, analyzed the data, wrote the paper, prepared figures and/or tables, reviewed drafts of the paper.

Russell J. Garwood analyzed the data, wrote the paper, prepared figures and/or tables, reviewed drafts of the paper.

Tristan Lowe performed the experiments, contributed reagents/materials/analysis tools, reviewed drafts of the paper.

Philip J. Withers contributed reagents/materials/analysis tools, reviewed drafts of the paper.

Phillip L. Manning conceived and designed the experiments, reviewed drafts of the paper.

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
