# Peer review of "Interpreting pathologies in extant and extinct archosaurs using micro-CT"

_PeerJ, doi:10.7717/peerj.1130_

## Round 0.1 · original submission · Major Revisions

· Academic Editor

Major Revisions

The manuscript has to be deeply improved, mainly in those aspects related to histopathology. Please, carefully considered all the recomendations made by the reviewers in the revised versión of it.

·

Basic reporting

The manuscript appears to meet all standards. I am not sure whether data sharing applies here, but I imagine the raw data files are several gigabytes in size and it may not be pratical to share these.

Experimental design

There appears to be some confusion about the selection of voltage, current and filter settings and how this relates to image contrast and detector saturation. Initially the text says that the voltage was adjusted to give 15-20% transmission (where does this figure come from; other published values are slightly lower, though the value is reasonable), but later it states that filters were used to achieve this. In general, filtering is applied to reduce the spectral width and so reduce beam hardening (prior to software correction). Common practice is to have a particular filter type and thickness according to the voltage, that is, each voltage setting would be associated with a particular filter (giving an opimal spectral width and intensity). The voltage/filter combination is then selected to give the correct transmission range. Current has no effect on transmission, only intensity. It is often limited due to the maximum power capacity of the anode. Detector saturation can be avoided by using shorter exposures or reducing the current (frame averaging can be used to increase the exposure without saturation). The limit of ~65000 counts would appear to be the analogue to digital converter limit, which would depend on gain settings and does not necessarily indicate detector saturation.
Total scan times would be useful, which help to give an idea of cost and viability of such studies.
Although TIFF is a well-recognised acronym, VGI might need spelling out and no reference is given for ImageJ.

Validity of the findings

Not being a biologist, I cannot comment on the valitidy of the pathalogical findings. However, the objective of the study is to demonstrate the efficacy of XMT to provide pathalogical information non-destructively and which would would not be achieved by other means. From the examples given, this is amply achieved.

Comments for the author

This is a valuable manuscript in demonstrating the capabilities of XMT in the field of palaeopathology.

Reviewer 2 ·

Basic reporting

1)The article does not display sufficient background to explain how it fits into the broader field of knowledge. The authors do not explain how XMT compares to the gold standard of bone histopathology, and do not compare to how lesions would have appeared histologically in comparison to the images shown.

2)The submission does not contain all results relevant to the hypothesis. As a paleopathology paper, the authors need to have two criteria 1) differential diagnoses for each specimen described- there cannot be a single diagnosis without comparison to excluded diagnoses and 2) there is no species specific comparison made, despite claims that this has been done.

Experimental design

1) While the research question was clearly defined, the investigation has not been conducted rigorously enough to meet expected standards. For a proof of concept paper for XMT evaluation of paleopathology, the authors do not put the concept in context with appropriate medical standards of histopathology. While I agree that bone morphology grossly is not ideal, the authors need to prove that XMT has enough detail to prove their point. As they do not use relevant references, and refer to a process that does not occur in reptiles and birds in multiple places, this cannot be proven.

Validity of the findings

1) The data is not robust and sound, because the underlying research and comparisons are not valid based upon evolutionary theory and valid species comparisons. The authors made little attempt beyond citing poultry articles to investigate the pathology of related species (e.g. zoological pathology or non-captive avian pathology).

Comments for the author

Comments:

Line 35: Change ‘pathologies’ to ‘lesions’.
Line 37: Change ‘paleopathologies’ to ‘ancient bone pathology’
Line 39: omit the word ‘just’.
Line 41: see correction from Line 37.
Line 41: ‘calli’ should be changed to ‘callus’
Line 46: perhaps ‘internal histologic superstructure’ rather than the less specific ‘morphologies’
Line 62: One could argue that the central canal in a Haversian system can be a lot smaller than 100um. How is this resolution sufficient to evaluate the sort of histologic detail that would truly allow you to describe histopathologic lesions?
A capillary at its largest is around 40um and an osteocyte is 5-20um. I think the authors must justify the limitations of resolution here.
Line 74: Rothschild is not a sufficient reference here as these publications actually do not sufficiently separate the diseases of taxa. The authors might consider citing a veterinary pathology text, or something along these lines.
Line 101: NHM S/1869.2.16.1 description. Osteomyelitis in archosaurs is not characterized by pus, therefore cloaca (a fluid pressure related phenomenon in mammals in which draining tracts form) do not exist. This is one of the dangers of citing sources which focus only on human pathology as the background of their work (e.g. Rothschild). As for the bumblefoot scenario- this process is driven by moist, damp, contaminated bedding by in large or poor sanitary conditions within a captive scenario. The authors are encouraged to consult zoologic medicine texts and journal articles to develop their differential diagnoses more appropriately in this section. Cooper’s books on raptor pathology, avian medicine textbooks are recommended. Also journal of zoo and wildlife pathology is a good source.

Line 123: BHI 6241- Again, the authors are encouraged to develop differential diagnoses based on relevant zoologic or veterinary literature. Having acknowledged the difference between hominid primates and other animal pathology, there must be followup on this comment. The sources noted are all human, even Rothschild, whose references are all human pathology related.

Line 136: This should be called ‘reactive arthritis’ requires the soft tissue pathology to evaluate and is ONLY a human condition. Additionally, the use of the term “Reiter’s” is sub-optimal given that he was an avid Nazi. Please develop species appropriate differentials.

Line 3033: BHI 3033, again, additional differential diagnoses should be developed. The authors should discuss the histologic appearance of osteomyelitis in archosaurs and how the lack of resolution might change their interpretation. Additionally, the authors should omit cloacae here and read Huchzermeyer’s paper on ‘fibriscess’, and possibly read Huchzermeyer’s ponderous book on Crocodile disease, as well as Mader’s Reptile Medicine and Jacobsen’s Infectious Diseases of Reptiles for more inspiration.

Line 163: Again, see earlier comments on ‘cloacae’ There need to be differential diagnoses developed here.

Line 184: I am pleased to see development of differential diagnoses here, however, again, having said that species relevant differential diagnoses would be developed from relevant literature, the literature cited is not archosaur specific, and the references underlying this literature is human. Please cite relevant zoologic and veterinary pathology literature of which is there is substantial availability.

Line 200-
The authors cannot justify these conclusions, as no comparison to histopathology, which is the gold standard is made for the conditions noted to the suprastructural information seen with this technology. In addition, the diagnoses cited are not underlain by evolutionarily relevant differential diagnoses. Therefore, while I agree, that this technology shows GREAT potential, I think that this paper requires considerable revision to show that potential.

---

## Round 0.2 · accepted · Accept

· Academic Editor

Accept

Authors have improved the manuscript following all the suggestions.

·

Basic reporting

The manuscript appears to meet all standards.

Experimental design

Following my previous review, the methodology has been clarified and all points addressed.

Validity of the findings

As mentioned previously, I am not able to comment on the validity of the pathalogical findings, but I do believe that the paper amply demonstrates the value of using XMT in this field.

Comments for the author

All points in the previous review appear to have been addressed.

Reviewer 2 ·

Basic reporting

Yes.

Experimental design

Yes. As said previously, it does meet these criteria.

Validity of the findings

The data is much improved with the authors revision of interpretations. I greatly appreciate the authors' revisions.

There are a couple of issues, which the authors could not have been aware of in their interpretations which will be mentioned below.

Comments for the author

Starting with line 135, I have some general comments.

One of the difficulties of the main veterinary/ zoologic vs. wildlife pathology literature is that some of the diseases that are noted are diseases of captivity rather than diseases of wild animals. Gout is seen in crocodilians mostly in captive animals and is mostly evident as visceral gout rather than the joint related gout seen in humans, which in and of itself is a lifestyle based disease from ingesting overly protein rich foods.

Several of these other metabolic bone diseases such as fibrous osteodystrophy, osteomalacia and rickets are seen primarily in captive rather than wild crocodilians- because they are tied to poor access to vitamin D, or low calcium in the diet which is not a problem seen in wild animals. I would be cautious in how this is applied to dinosaurs. Metabolic bone disease is the greatest challenge in captive reptiles and so this dominates the literature for vet med.

Also- there is a great paper on avian neoplasms from Victoria if you are looking for some additional animal tumor information.

I cannot state enough how much I appreciate your revisions.

Sincerely,

Ewan Wolff, PhD, DVM
Resident, Small Animal Medicine
Purdue University

Research Associate, Museum of the Rockies

Affiliate Assistant Professor, Montana State University